# Immune Priming Triggers Cell Wall Remodeling and Increased Resistance to Halo Blight Disease in Common Bean

**DOI:** 10.3390/plants10081514

**Published:** 2021-07-23

**Authors:** Alfonso Gonzalo De la Rubia, Hugo Mélida, María Luz Centeno, Antonio Encina, Penélope García-Angulo

**Affiliations:** Área de Fisiología Vegetal, Departamento de Ingeniería y Ciencias Agrarias, Universidad de León, E-24071 León, Spain; algor@unileon.es (A.G.D.l.R.); hmelm@unileon.es (H.M.); mlcenm@unileon.es (M.L.C.); a.encina@unileon.es (A.E.)

**Keywords:** bean, *Pseudomonas*, cell wall, plant defense, disease resistance, 2,6-dichloroisonicotinic acid, INA

## Abstract

The cell wall (CW) is a dynamic structure extensively remodeled during plant growth and under stress conditions, however little is known about its roles during the immune system priming, especially in crops. In order to shed light on such a process, we used the *Phaseolus vulgaris*-*Pseudomonas syringae* (Pph) pathosystem and the immune priming capacity of 2,6-dichloroisonicotinic acid (INA). In the first instance we confirmed that INA-pretreated plants were more resistant to Pph, which was in line with the enhanced production of H_2_O_2_ of the primed plants after elicitation with the peptide flg22. Thereafter, CWs from plants subjected to the different treatments (non- or Pph-inoculated on non- or INA-pretreated plants) were isolated to study their composition and properties. As a result, the Pph inoculation modified the bean CW to some extent, mostly the pectic component, but the CW was as vulnerable to enzymatic hydrolysis as in the case of non-inoculated plants. By contrast, the INA priming triggered a pronounced CW remodeling, both on the cellulosic and non-cellulosic polysaccharides, and CW proteins, which resulted in a CW that was more resistant to enzymatic hydrolysis. In conclusion, the increased bean resistance against Pph produced by INA priming can be explained, at least partially, by a drastic CW remodeling.

## 1. Introduction

In nature, plants are usually under biotic stress caused by different pathogens, making their survival difficult [1]. To combat attackers, plants have developed a sophisticated immune system, whose activation usually produces a decrease in yield and fitness [2], and therefore it must be tightly regulated. In this immune system, the plant cell wall (CW) is the first physical barrier to prevent the pathogen from progressing into the cell, having the capacity for remodeling the architecture of its components and their physicochemical properties [3,4,5].

Plant CW is a dynamic structure which surrounds every plant cell, determining their shape and providing mechanical support to the protoplasts to counteract the turgor pressure. Elongating cells are walled by the primary CW, a thin layer in continuous change, mainly composed of a complex matrix of cellulose, pectins, and hemicelluloses, and structural glycoproteins [6,7]. Cellulose is the main CW scaffold, which is composed of numerous β-(1-4)-glucans tightly packed to form microfibrils, very resistant to enzymatic hydrolysis. The hemicelluloses and pectins, commonly named as matrix polysaccharides, are a group of several distinct polysaccharides [7]. Pectins are a group of complex acid polysaccharides that are divided in three main domains by their occurrence: homogalacturonan (HG), rhamnogalacturonan I (RG-I), and rhamnogalacturonan II (RG-II). The proportion of these pectin domains depends on species, tissue, and even cell type, with the presence of minor pectic domains such as apiogalacturonan or xylogalacturonan. The binding among these pectins results in complex macromolecular structures which form a hydrated gel phase where cellulose and hemicelluloses are embedded [8]. They are mostly extracted from the CW by means of calcium chelators or hot water treatments. Hemicelluloses usually bind to the surface of the cellulose microfibrils, often bearing short side branches, and are typically extracted under alkali incubation. Dominant hemicelluloses in primary CW are xyloglucan (XG), xylan, or arabinoxylan (AX) depending on plant species [9]. Accompanying polysaccharides, structural proteins such as hydroxyproline-rich glycoproteins (HRGP) or arabinogalactan proteins (AGPs) are widespread in the CW matrix [10,11]. When the cell stops growing, a strengthened secondary CW, mainly composed of cellulose, hemicelluloses, and lignin, is deposited between preexisting primary CW and the plasma membrane [12,13].

As plant morphogenesis ultimately depends on CW properties, this structure needs to adapt to any cellular change at any developmental stage along the entire life of a plant [10]. These CW processes are tightly regulated to control the metabolism responsible of CW remodeling. Firstly, the synthesis loci differ depending on the CW polymers. Cellulose is synthesized at the plasma membrane [14], while matrix polysaccharides are deposited throughout the secretory apparatus [15,16]. Secondly, the properties that these polymers initially could confer to the matrix can be changed once they are deposited, by homo and/or hetero-crosslinking and by modifying the pattern and/or degree of backbone substitution. For instance, HG is synthesized in Golgi apparatus with most of its acidic carboxyl groups methyl-esterified, however later on, by removing the methyl groups, ionic bridges can be formed by calcium ions, establishing a network among HG chains [8]. RG-I and RG-II, which are pectins decorated with neutral sugars chains, are complex polysaccharides whose abundance, cross-linking and substitutions are also regulated in muro, depending on the development stage [8,17]. To gain resistance, RG-I backbone can be enlarged, rendering a higher molecular weight of the polysaccharides. This can be also achieved by cross-linking throughout borate diesters between the RG-II domains [17]. On the other hand, typical modifications of dicotyledonous hemicellulosic glycans are the presence of fucose or galactose residues attached to the XG backbone, which is important during the cell elongation [18]. In the same way, extensins, which are HRGP, play a crucial role in the structural and compositional CW plasticity, by forming intra- and intermolecular cross-links [19].

CW remodeling serves to strengthen the CW, maintaining the plasticity to react to environmental changes, being especially important under a pathogen attack. The role of the CW components in defense has been attested in different works carried out with several mutants mainly of the model species *Arabidopsis thaliana* (Arabidopsis), in which a specific CW alteration translated into resistance or vulnerability against different diseases [20]. For example, the impairment of cellulose synthesis in an isoxaben-resistant mutant (*ixr1*, also known as *cev1*) resulted in an enhanced resistance to fungal pathogens by means of constitutive activation of some defensive pathways [21]. Similarly, Arabidopsis mutants unable to synthesize cellulose in the secondary CW, as *irx1-6*, showed resistance against the fungus *Plectosphaerella cucumerina* and the bacterium *Ralstonia pseudosolanacearum* [22]. The impairment in the callose deposition, observed in constitutively activated cell-death (*cad1-3*) Arabidopsis mutants, produced an increment to *Pseudomonas syringae* susceptibility [23]. In the case of lignin, the overexpression of *WRKY1* gene in rice, which increased lignification and resulted in major resistance against *Verticillium dahlia* [24]. By contrast, downregulation of the genes involved in lignin biosynthesis (hydroxycinnamoyl-CoA shikimate quinate hydroxycinnamoyl transferase -HCT- or cinnamoyl-CoA reductase 1 -CCR1-), with alteration in lignin content and composition, triggered defense responses in Arabidopsis [25]. Interestingly, the role of polysaccharide feruloylation in monocot *Brachypodium* and dicot Arabidopsis was also demonstrated by the transgenic expression of a fungal feruloyl esterase in both species, which reduced mono and/or dimeric ferulic acids, resulting in major susceptibility to *Bipolaris sorokiniana* and *Botrytis cinerea*, respectively [26].

Similar effects were found when matrix polysaccharides were altered. Arabidopsis mutants repressed in glucuronate 4-epimerases genes (*gae1* and *gae6*) had an impairment in galacturonic acid and showed less resistance to *P. syringae* pv. *maculicola* and *Botrytis cinerea* [27]. In addition, Arabidopsis response regulator 6 mutant (*arr6*) was enriched in pectins and showed more resistance to *P. cucumerina* but more susceptibility to *R. pseudosolanacearum* [28]. Reductions in the content of galactomannans and in the decoration of xyloglucan, observed in immutans (*im*) variegation of Arabidopsis, involved susceptibility to *P. syringae* [29]. Finally, with respect to CW proteins, the reduced residual arabinose-2 (*rra2*) mutant, defective in extensin arabinosylation, could limit oomycete colonization in Arabidopsis roots [19]. Other CW enzymes known to participate in pathogenesis are pectin modified enzymes, such as pectin methylesterases (PME), which take part in the demethylesterification of HG. An example is the infection of *P. syringae pv. macuolicola* in Arabidopsis, which leads to an increment of PME activity [27]. Thus, *pme17* Arabidopsis mutants showed an increase in susceptibility to *B. cinerea* [30]. In the same way, PME inhibitor (PMEI) mutants such as *pmei10*, *pmei11*, and *pmei12* lead to immunity against *B. cinerea* [31].

Apart from mutants, another strategy consists in comparing resistant and susceptible wheat breeding lines to *Fusarium graminearum*. It was revealed that resistance could be due to CW changes such as a higher content of lignin, a higher degree of arabinosylation on xylans, and a higher degree and different pattern of methylesterification on pectins [32].

All these works were based on CW modified mutants or breeding lines that showed susceptibility or resistance against pathogens, but their CW was already modified on these genotypes, so the level of susceptibility was determined by a preformed structure that in some mutants involved defensive pathways constitutively activated [20]. However, much less is known about CW remodeling at a physiological stage after pathogen infection. In this sense, most studies have focused on interactions with necrotrophic fungi or insects, which produces mechanical damage in the CW by the release of CW degrading enzymes (CWDEs) [26,32]. The main documented responses so far are the local deposition of CW materials, known as papillae or the cross-linking among components of the matrix, in order to hinder the access of the pathogens to the protoplast [22,31,32]. Recently, several transcriptomic and proteomic analysis would indicate that CW remodeling is a widely extended process in nature [33,34,35,36], however the detailed characterization of the changes that take place in muro after bacteria attack remain poorly characterized.

Among other defense layers, it has been proposed that cellular defense responses can be activated, but stronger, faster, and more efficiently, by means of the application of different treatments, a phenomenon known as immune priming [37]. However, it is not known whether, somehow, this immune priming conducts to the strengthening of the CW, or the immunity triggers a remodeled CW capable to cope with a second attack from pathogens. Over the past decades, several stimuli, compounds, or abiotic stresses have been studied in order to produce priming [37,38,39,40,41]. This new state is reached by changes at physiological, transcriptional, metabolic, and epigenetic levels, which can hold priming throughout the entire life of a plant and can even be inherited by further generations [42,43]. Among other compounds proposed to produce this long-term effect, it is the functional salycilic acid (SA) analogue 2,6-dichloroisonicotinic acid (INA), the first synthetic compound to induce defense responses in laboratory [39]. Recent studies have proposed that INA enhances basal disease resistance in common bean (*Phaseolus vulgaris* L.) by epigenetic changes that can even be inherited [41,43]. However, although it has been described that the priming activator (R)-β-homoserine increased Arabidopsis CW defense against *H. arabidopsidis* by enhanced callose deposition [41], there are no studies that focus on CW remodeling after INA priming.

In this study, *Riñón* common bean was used. It is a widely cultivated variety in León (Spain), under the protected geographic identification (PGI) “Alubia de la Bañeza-León”. This variety is attacked by the biotrophic gamma-proteobacteria *Pseudomonas syringae* pv. *phaseolicola* (Pph), causing halo blight disease [44,45,46], which causes yield losses of up to 45%, with its main symptoms being general chlorosis, stunting and distortion of growth [45]. Until now, disease control has mainly consisted in growing healthy seeds every season or replacing the susceptible varieties with others more resistant. This latter alternative is not suitable when the susceptible variety cultivated has gastronomical and economic interest, such as *Riñón* [45]. *Riñón* common bean has been previously described as a Pph-susceptible variety [44], although the reasons behind this susceptibility are unknown. A recent work from our group revealed that this bean variety is able to perceive the presence of Pph but it is unable to produce an effective and fast defense response, at least in part because of the lack of a quick SA peak production [47]. This is the reason why the use of SA analogue, such INA, could help in the defense process against Pph. On the other hand, there is no knowledge about CW remodeling after the Pph infection and it becomes relevant because Pph is a CWDEs producer [48].

Taking all of this into account, the aims of the present work were to know: (1), if a susceptible bean variety can remodel its CW after Pph inoculation, (2) if the application of a priming compound, such as INA, has effects in CW remodeling, and finally, (3) if these CW changes increased bean resistance against Pph. For these purposes, 15-day-old bean plants were pretreated or not with INA, and 1 week after they were inoculated or not with Pph. One-week post infection, the plants were collected and their CWs were extracted for the subsequent analyses.

## 2. Results

### 2.1. INA Reduced Bean Susceptibility to Pph and Increased Flg22-Triggered Response

To determine whether the INA application was able to prime bean plants defense as it was described before [42,49], the phenotypical symptoms after Pph inoculation were recorded. Plants were grown in vitro, a system in which infection can be forced through high pathogen concentration and humidity and without other environmental stresses [50,51]. When plants were 2 weeks old, the experimental conditions were established as follows: bean plants pre-treated with INA (INA) or not (Mock) were inoculated with Pph (INA + Pph, and Mock + Pph, respectively) (Appendix A). The observed disease symptoms were similar to those previously described [45]. Mock plants grew without symptoms (Figure 1A), as their foliar color and development corresponded with the normal growth of non-stressed plants. By contrast, Mock + Pph-inoculation triggered general chlorosis, greasy appearance on leaves, and the development of necrotic tissue areas, as shown in Figure 1B, indicating the susceptibility of these plants against the pathogen. INA did not cause visible stress symptoms compared to Mock (Figure 1C). Interestingly, foliar damage produced by the Pph inoculation was reduced considerably in INA + Pph plants (Figure 1D). In this case, chlorosis was restricted to foliar margins, and no greasy leaves appeared. Therefore, compared to Mock + Pph, INA + Pph plants were more resistant to Pph, and showed a statistically smaller lesion area (Figure 1E), and interestingly showed small necrotic spots in interveinal areas, which could be the result of a hypersensitive response (highlighted area in Figure 1D), as has been observed in other bean varieties resistant to Pph [52].

In view of the results obtained, whether INA priming was able to modify the ability of Riñón variety to trigger a general defense response was investigated. For this purpose, reactive oxygen species (ROS) production in bean leaf disks was monitored by a peroxidase/luminol-based assay (See Appendix A). This method is commonly used to reveal whether pathogen- or host-derived ligands are able to trigger early immune hallmarks [53]. In our experimental system, the INA application did not trigger a ROS production compared to mock (Figure 2). However, INA pre-treatment produced the highest ROS peak after the addition of the bacterial-derived peptide flg22 (which may mimic Pph inoculation). This suggested that INA did not induce a typical immune-associated fast response such as ROS production but primed the bean cells for more intense defense responses.

### 2.2. Pph Inoculation and INA Priming Produce Different CW Fingerprinting Than That Observed in the Mock

Several recent evidence support a more prominent role of the CW in plant immunity than previously believed, although the knowledge about plant CW remodeling after pathogen infection and/or immune priming is scarce [4,5,22,26]. With this in mind, CWs from Mock, Mock + Pph, INA and INA + Pph bean plants were extracted. Characterization of whole CWs began with the non-invasive technique FTIR-spectroscopy, used to obtain global fingerprints of the CW [54]. The FTIR spectra obtained were clustered in a Principal Component Analyses (PCA), which separated the treatments in two groups (Mock and the rest of the treatments) according to dimension 2, which explained 29.43% of variance. In order to predict the putative CW components contributing to such segregation, the wavenumbers whose values had a higher correlation with dimension 2 were extracted and summarized in Table 1 together with their associated CW components. Several wavenumbers were associated with RG-I, galactan and xyloglycan, (1148 and 1152 cm^−1^), pectins (1232 and 1244 cm^−1^), uronic acids (1616–1628 cm^−1^), or with arabinogalactans (1156 cm^−1^). In addition, the wavenumbers related to cellulose (1160 and 1164 cm^−1^) and phenolics (1632 and 1720 cm^−1^) were also found. Next, difference spectra against mock were calculated for each condition (Figure 3B). As a result, the profile between 1800 and 1170 cm^−1^ approximately, which referred among others to wavenumbers regarding uronic acids and pectins, was similar among all conditions. By contrast, the region from 1170 to 800 cm^−1^ associated to RG-I, Galactan, Xyloglycan, Arabinogalactan and Cellulose (Figure 3B, Table 1) only changed after INA pre-treatment (with or without further Pph infection) when compared to mock.

### 2.3. INA Priming Induced Quantitative Changes in CW Polysaccharides Not Observed after the Pph Inoculation

To deepen into the CW changes after the Pph inoculation and the INA priming, the cellulose content was measured in crude CWs. It should be taken into account that cellulose, a major CW component, has been described to participate in defense and remodeling [20]. Besides, our results indicate that cellulose contributes to discriminate among treatments after PCA of FTIR CW spectra (Table 1). Updegraff methodology confirmed that the INA pre-treatment resulted in an increased crystalline cellulose content in both INA and INA + Pph plants, which could indicate a CW reinforcement after priming, due to the fact that the simple Pph inoculation of Mock did not alter the cellulose content significantly (Figure 4).

In order to explore the possibility of a CW reinforcement occurring after INA priming, a sequential CW extraction of matrix polysaccharides was performed. During the CW fractionation, pectic polysaccharides are commonly extracted in CDTA and Na_2_CO_3_ fractions, while hemicelluloses are mainly extracted by incubation with different alkali (normally KOH or NaOH) concentrations, which extract hemicelluloses slightly (KI fraction) and tightly (KII fraction) bound to cellulose and/or cross-linked (Cosgrove, 2018). The following fractions recover polysaccharides tightly bound to cellulose (SnCR) and amorphous cellulose (TFA) [61], respectively.

The Pph inoculation only had an effect in pectic fractions, producing an increment of neutral sugars and uronic acids in CDTA fraction as well as an increment of uronic acids in Na_2_CO_3_ fraction compared to mock (Figure 5). However, the INA pretreatment produced dramatic changes in the neutral sugar distribution, by decreasing its content in CDTA, Na_2_CO_3_ and SnCR fractions, and significantly increasing it in TFA fraction (Figure 5A). The uronic acid content diminished in CDTA by INA pretreatment, but increased in pectic (Na_2_CO_3_) and hemicellulosic (KII) fractions, as well as in the SnCR fraction (Figure 5B). These results show a clear shift of polysaccharides from the CDTA fraction to others where extracted polymers were linked to the matrix more strongly, thus suggesting changes in their composition and/or structure. Interestingly, these CW changes observed in INA were similarly observed after the Pph inoculation (INA + Pph).

Thereafter, the fractions were subjected to acid hydrolysis, and neutral sugar composition was determined by gas chromatography (Figure 6). After the INA pretreatment, no neutral sugars were detected in CDTA and Na_2_CO_3_ fractions, probably due to an increment in the uronic acid to neutral sugar proportion in these fractions (Figure 5), which makes the neutral sugar quantification difficult [62]. In these pectin-enriched fractions (Figure 6A,B), the Pph inoculation produced increments in rhamnose (Rha) and arabinose (Ara) concentrations, most likely arising from RG-I. Additionally, Ara and galactose (Gal) also increased in SnCR and TFA fractions in Mock + Pph (Figure 6E,F), pointing also to an enrichment in arabinogalactans.

In the INA pretreatments a displacement of Rha, Ara, xylose (Xyl), Gal, and glucose (Glc) monosaccharides, from KI, KII, and SnCR fractions to the TFA fraction, was observed (Figure 6C–F), indicating that the polysaccharides implicated in the CW remodeling after priming could be RG, XG, and/or xylans, more tightly linked to the matrix. A decrease in the mannose (Man) content was observed in KI and SnCR fractions after the INA pretreatments compared with Mock. In addition, the Glc increase found in the TFA fraction after the INA pretreatment (Figure 6F) could be related with the higher cellulose content found in crude CWs for those conditions (Figure 4) or even to an increment in other beta-linked-glucans able to reinforce the CW structure such as callose [63,64].

Finally, when plants were inoculated with Pph after priming (INA + Pph), no substantial changes were observed compared with INA, except an increment in Gal and Glc in KI (Figure 6C), and in Ara, Xyl, and Glc in KII fractions (Figure 6D), which could derive from hemicelluloses

### 2.4. INA Priming Increased the Average Molecular Weight of Polysaccharides in All CW Fractions

Additional information about relative mass distribution and average molecular weights (M_w_) of polysaccharides in the CW matrix was obtained by gel permeation chromatography [65] (Figure 7). Mock + Pph involved a M_w_ increase of all fractions compared to Mock (Figure 7), except in KII, being the most notorious increase in the Na_2_CO_3_-pectic fraction (Figure 7B), where polysaccharide dispersion was higher. Interestingly, the INA pretreatment was always associated with a remarkable increase in the estimated M_w_ of all polysaccharide populations analyzed. This shift was particularly noticeable in hemicellulosic fractions in which a 14-fold (KI fraction) and 6.5-fold increase (KII fractions) was found (Figure 7C,D). In these cases, INA + Pph had nearly negligible impact on M_w_ compared to INA. These results obtained after INA priming pointed to the possibility of CW remodeling by an extensive polymer cross-linking. Interestingly, the elution profile shown two populations of polysaccharides after the INA pretreatment in fractions CDTA (Figure 7A), KI (Figure 7C) and KII (Figure 7D): (i) relative to lower M_w_, which corresponded with Mock and Mock + Pph population, and (ii) of higher M_w_, that did not appear in those not treated with INA.

### 2.5. Qualitative Epitope Changes in CWs Produced by the Pph Inoculation and INA Priming

As the Pph inoculation, as well as INA priming, involved a displacement of polysaccharides populations extracted across all the fractions (Figure 5 and Figure 6), and the increment in M_w_ of these polysaccharides (Figure 7), an epitope characterization analysis of these CWs was carried out (Figure 8). Therefore, a screening of CW qualitative changes was performed by immunodot assays (IDAs) against different CW proteins such as extensins (recognized by LM1 monoclonal antibody [66]) and arabinogalactan proteins (AGPs, recognized by LM2 [9]). HGs with different degree and pattern of methyl esterification were screened by means of JIM5 or JIM7 antibodies, which probe to low and high methyl esterified HGs respectively [67]. The β-1,4-galactan and α-1,5-arabinan side chains of RG-I were also probed by using LM5 or LM6 antibodies, respectively [9]). In addition, hemicellulose epitopes were analyzed with LM11, which detects xylans and AXs [68]), and LM15, which labels non-fucosylated XG [69]).

The upper side of the heatmap shows the relationship among antibodies binding profiles. Xylan/arabinoxylan (LM11) and extensin (LM1) antibodies clustered together (cluster A) and had less CW epitopes than pectic and hemicellulosic antibodies, which clustered in the other group (cluster B). Within group B, HG antibodies (JIM5 and JIM7) had the more intense binding profile (cluster B.1), while RG-I (LM5 and LM6), XG (LM15), and AGP (LM2) antibodies grouped in an independent sub-cluster (B.2).

Attending to the left part of the heatmap, two clusters were observed: cluster C, which included different fractions from INA-pretreatments (INA and INA + Pph), and cluster D, in which Na_2_CO_3_, KI, and SnCR fractions from Mock and Mock + Pph plants were included. Interestingly, Na_2_CO_3_ and KI fraction always grouped together although in different clusters depending on the treatment (cluster C.1a for INA and INA + Pph, and cluster D for Mock and Mock + Pph), whereas CDTA and KII fractions were clustered independently in cluster C.1b and cluster C.2, respectively. The clustering analysis after IDAs revealed a similar epitope profile for CDTA and 4N KOH (KII) extracted polysaccharides independently of the treatment. Besides, INA and INA + Pph linked to a differentiated labelling pattern in Na_2_CO_3_ and 0.1N KOH (KI) extracted polysaccharides when compared to Mock or Mock + Pph. As expected, in Mock plants the pectin fraction extracted with CDTA was enriched in HG with high and low degree of methyl esterification (JIM7 and JIM5) and RG-I (LM5 and LM6), but also non-fucosylated XG and AGPs were found. Mock + Pph produced only a slight increase in HG methylation degree (JIM7) in the CDTA fraction. However, INA and INA + Pph also increased the presence of extensins (LM1) and AGPs (LM2), and decreased galactan side chains of RG-I (LM5) and XG (LM15) epitopes, mainly in INA + Pph.

In Mock, CW polysaccharides extracted in Na_2_CO_3_ and KI fractions were enriched in RG-I (LM5 and LM6), HG (JIM5 and JIM7), XG (LM15) and AGPs (LM2), but the Na_2_CO_3_ fraction had only a small proportion of extensin epitopes (LM1), and KI had only a small proportion of xylan epitopes (LM11). Mock + Pph did not substantially change the epitope distribution in Na_2_CO_3_ and KI fractions. However, the epitope profile of these fractions dramatically changed after INA pretreatment, mainly due to the increase in extensin (LM1) and HG (JIM5 and JIM7) epitopes, especially in the Na_2_CO_3_ fraction after the Pph inoculation (INA + Pph).

The KII fraction was mainly composed of XG (LM15) and xylans (LM11) in Mock, although HG (JIM5 and JIM7), RG-I (LM5 and LM6) and AGP (LM2) epitopes were also detected. Mock + Pph itself did not substantially affect this profile, but the INA pretreatment produced a pectin increment, mainly HG with a high degree of methylation (JIM7) and RG-I (LM5 and LM6) and an increase in extensin epitopes.

Finally, the SnCR fraction was more variable in composition depending on the treatment. In Mock, SnCR was composed of pectins (RG-I and HG), hemicelluloses (XG and Xylans), and AGP. However, in Mock + Pph a decrease in hemicellulosic (LM11 and LM15) and HG (JIM5 and JIM7) epitopes was observed. The composition of this fraction was also different between INA pretreatments. The INA pretreatment increased, as in other fractions, the presence of HG (JIM5 and JM7) and extensin (LM1) epitopes, but the Pph inoculation (INA + Pph) also increase the abundance of hemicellulose (LM15) and AGP (LM2) epitopes compared to Mock.

### 2.6. INA Priming Prevented Enzymatic Digestibility of CWs

The CW changes observed after the Pph inoculation, and especially after the INA priming, demonstrated CW remodeling events. In order to know whether such CW remodeling confers more resistance to pathogen attack, especially taking into account that Pph is able to produce CWDE, a time course of sugar release after enzymatic hydrolysis of crude CWs was carried out (Figure 9). In this assay, no significant differences were found when both, sugar yield and sugar release kinetics obtained from Mock + Pph and Mock CWs were compared. However, INA pre-treatments induced a slower release of total sugars during the first half of the time course. Samples from INA + Pph did not reach sugar release levels of Mock after 24 h. In summary, the INA priming induced a CW reinforcement, making the structure more resistant to enzymatic hydrolysis.

## 3. Discussion

In nature, plants must overcome multiple stresses, such as those induced by pathogens, by activating several defense mechanisms. The CW is the first plant defensive layer against intruders. Plants are able to rearrange this structure on demand at different developmental stages. The CW can be reinforce by modifying the proportions of its main components, and/or by changing the type and extent of cross-linking among CW polymers [4]. To gain insight into the pathogen and priming capacity to reinforce the CW, the remodeling of CWs after the Pph inoculation and INA priming in the Pph-susceptible common bean variety *Riñón* was studied in this work.

Previous studies have suggested the ability of INA, as a SA structural analogue, to trigger the immune system in common bean [40,42,49]. To confirm the priming activation in *Riñón* common bean, plants of this variety previously treated or not with INA were inoculated or not with Pph (Appendix A). By following that approach, it was confirmed that plants pre-treated with INA showed less halo blight symptoms [45], as the chlorosis and greasy appearance observed after the Pph inoculation decreased (Figure 1). The fact that these symptoms were constrained to damage at the foliar margins confirmed that INA pretreatment prior to the Pph inoculation protected *Riñón* bean plants from halo blight disease, as it was previously observed in other varieties [42,49]. As priming suggests, this observation may indicate that cell defensive responses have already been activated, and consequently, a stronger defensive mechanism is expected [37,41]. In order to confirm such hypothesis, the ROS burst triggered by peptide flg22 [70] was evaluated in bean leaf disks previously incubated or not with INA. As shown in Figure 2, a more intense ROS signal after flg22 triggering was detected in those plant disks previously treated with INA, which was in line with the priming concept, as it primed the plant cells to develop a more vigorously defense response somehow [71,72,73,74]. Ramirez-Carrasco et al. (2017) [42] proposed that this SA analogue produces epigenomic changes in bean that can even be inherited [43,49].

The presence of a CW that shields plants from pathogen invasion is a common resistance mechanism to disease among all plant cells. Several recent works have demonstrated a more relevant role of the CW than initially expected mediating such resistance. Most of them have focused on the study of mutants with a great variety of CW alterations, which have resulted in differential resistance or susceptibility against plant pathogens [4,20]. Other works have studied the role of elicitation with conserved CW-derived molecules from the pathogen, or from the plant itself, which once perceived the pathogen and triggered different immune responses [75,76]. However, the knowledge about CW remodeling after immune priming or bacterium infection is scarce. Here, we studied whether Pph inoculation and/or INA priming in common bean could promote a CW remodeling which would end up in the enhanced protection against halo blight disease, as shown in Figure 1.

The study of CW remodeling in this work began with the non-invasive technique FTIR spectroscopy of crude CWs, which resulted in the discrimination of FTIR spectra of Mock + Pph, INA and INA + Pph samples from the Mock samples (Figure 3). Dimension 2 of the PCA applied, which explained almost 30% of variance, was mainly related to wavenumbers associated with pectins, cellulose, and phenols (Table 1). The subsequent analysis of the polysaccharide content revealed changes in the amount of cellulose (Figure 4), matrix polysaccharide distribution (Figure 5), monosaccharide composition (Figure 6), and changes in the inmunoprofile of pectins (Figure 8).

When bean plants were inoculated with Pph, no statistical differences were found in cellulose content (Figure 4), in contrast with previous findings from other pathosystems [3]. However, our results point to both quantitative and qualitative changes in pectins and hemicelluloses after the Pph inoculation (Figure 5 and Figure 6). The total amount of pectins (CDTA and Na_2_CO_3_ extracted polysaccharides) increased upon the Pph inoculation (Figure 5). Indeed, the higher HG methylation degree observed, as revealed by immunolabeling in these fractions (Figure 8), would suggest a HG synthesis increase after the infection, as HG methyl-esterification only occurs in Golgi apparatus [77,78,79], and/or a decrease of pectin methyl esterase activity after the Pph infection, as this is altered in other pathosystems [27,80]. Besides, the Ara and Xyl variation in the KII fraction suggest an increment of AX in tightly cross-linked hemicelluloses. The M_w_ of pectins (particularly those Na_2_CO_3_ extracted) and loosely crosslinked hemicelluloses (KI fraction) raised after the Pph inoculation. A putative increase in the length and/or number of arabinan and galactan side chains of RG-I was also reported.

Several studies show how plants modify their pectin methylation degree or substitution pattern in order to avoid microbe colonization [78,81]. It is well-described that HG demethylation, as it is necessary to establish links between HG chains through Ca^2+^ bonds to form the denominated egg-box complexes [82], participate in biotic resistance [30,31]. This type of link could explain the M_w_ increase observed for pectic fractions (Figure 7) [83,84]. These changes in pectic polymers could be not enough as defensive mechanisms, as it was observed phenotypically (Figure 1). In fact, the CW degradability profile was similar in Mock + Pph and Mock (Figure 9). Pph is characterized, as other pathogenic microbes, by the production of CWDE that are released in the apoplast, similar to those used in the CW degradability assay [4,48]. This could be an explanation to the symptoms observed caused by the disease.

The Glc increase in the TFA fraction may reflect the accumulation of non-crystalline cellulose after the INA treatment (Figure 4). INA, as structural SA analogue, could stimulate cellulose synthesis. This polysaccharide is precisely regulated by growth factors, and previous studies have suggested the relation between SA and cellulose accumulation [85].

Alternative explanations for the Glc increase in the TFA fraction would be the accumulation of callose and/or the enrichment in a XG population tightly bound to cellulose. Both explanations seem plausible as the protective role of callose against Pph attack [86] and the function of XG by strengthening CW structure [63,64] have been previously reported. The CW fractionation of plants pre-treated with INA showed a noticeable displacement of matrix polysaccharides among fractions.

Regarding Mock and Mock + Pph conditions, neutral sugars were barely detected in CDTA and Na_2_CO_3_, decreased in SnCR, and increased in TFA (Figure 5). Specifically, the displacement of monosaccharides to the TFA fraction would suggest that RG-I, XG and xylans are the polysaccharides involved (Figure 6). A similar displacement from CDTA to Na_2_CO_3_, KII and SnCR could have occurred regarding uronic acids (Figure 5B). These changes indicated that polysaccharides become strongly attached to the matrix. Moreover, marked changes in polysaccharide size (Figure 7) and inmunoprofile in epitopes for each fraction (Figure 8) would suggest INA-dependent changes in the CW structure. The increase in glycan M_w_ observed for all fractions in INA treatments compared to Mock (Figure 7) could be explained by changes in cross-linking detected in CW epitopes (Figure 8), which is interesting when the sugar content decreased in CDTA and did not change in KI and KII (Figure 5). Galactan or arabinan side chain substitutions in RG-I decreased in CDTA but increased in Na_2_CO_3_ after the INA pretreatments. Additionally, AGPs or XG diminished after the priming, but all fractions obtained from plants pre-treated with INA showed an increment in HG and extensin epitopes (Figure 8). As being part of the same molecule [87,88], the changes observed in RG-I could have indirectly rendered a HG more attached to the matrix and, consequently, the detection of HG in fractions such as KI and KII, in which the presence of pectins is not usually abundant [89,90].

The more distinctive fact of CW fractions from INA treatments compared with those obtained from Mock and Mock + Pph CWs, was the increment in extensin epitopes, which could participate in matrix remodeling, maintaining the plasticity. Extensins are hydroxyproline-rich glycoproteins which interact with other CW components such as pectins, participating in the architecture, the structural organization and the strengthening of the CW [91,92,93]. Therefore, these proteins are involved in polysaccharide cross-linking [92,93]. Interestingly, extensin overexpression produces a decrease in the susceptibility of Arabidopsis to *Pseudomonas syringae* DC3000 [94]. In line with this, Arabidopsis plants with a reduced wall-extensin content are less resistant to *Botrytis* [26]. In addition, it is well known that some extensin genes are induced in *Arabidopsis* and *Nicotiana* by SA [95,96,97], and INA, as a SA analogue, could produce a similar effect in bean.

Taken together, all these changes suggest a remodeling of the CW architecture, which was finally confirmed by the enzymatic digestion assays, as CWs extracted from plants treated with INA showed a higher resistance to enzymatic hydrolysis (Figure 9). These differences, which were especially noticeable at early times, could be explained, at least partially, by a CW reinforcement after priming. The inoculation of the pathogen after the INA priming meant slight CW modifications, which was also reflected in a similar CW degrading profile upon enzymatic digestion. As previously reported [98], the CW cross-linking and strengthening could hamper pathogen CWDEs activity and confer biotic resistance, which would explain the slight lesions produced by halo blight after the INA pre-treatment in this variety (Figure 1). INA pre-treated plants were more resistant to Pph, and showed small necrotic spots, which resembled those produced during the HR response (Figure 1D). In resistant plant species, SA accumulates in the infection site at early stages of the immune response, which leads to HR [99]. However, previous studies showed that *Riñón* variety was not able to produce the early SA peak required after the Pph attack to induce HR [47]. Therefore, INA could replace the role of SA by increasing the defense signal at early stages after Pph infection in this variety.

To summarize, the Pph inoculation modified the common bean CW to a much lower extent than INA-priming. It did not involve an increase in cellulose content, but increased pectins (HG and RG) in pectic fractions and hemicelluloses (AX) in KII fraction. The M_w_ of polysaccharides increased most of all in Na_2_CO_3_ fraction, but the CW was as vulnerable to enzymatic hydrolysis as Mock. By contrast, INA priming triggered a drastic CW remodeling, by increasing the cellulose content, displacing the matrix polysaccharides among fractions, and increasing the M_w_ of polysaccharides extracted in each fraction. This could be related to an increment in pectins (HG and RG) and extensins in all fractions, which supported a more extensive cross-linking that resulted in a CW more recalcitrant to enzymatic hydrolysis. The Pph inoculation after INA priming did not modify substantially this CW remodeling and the resultant CW was as resistant to enzymatic degradation as INA without inoculation. Therefore, INA-priming-phenotypes regarding more intense ROS production after flg22-elicitation and increased resistance against Pph infection, could be explained, at least partially, by the CW remodeling observed. Future work to unveil the specific links between CW remodeling and disease resistance will pave the way to design novel crop protection strategies based on such knowledge.

## 4. Materials and Methods

### 4.1. Plant Material

Seeds of common bean *P. vulgaris* L. cultivar *Riñón*, from the Protected Geographical Identification *La Bañeza-León* region (Spain), were sterilized with 70% (*v*/*v*) ethanol for 30 s and 0.4% (*w*/*v*) NaClO for 20 min, prior to washing with sterile water. Germination took place under in vitro conditions using glass jars (946 mL of volume) with universal sterilized substrate (Blumenerde, Gramoflor, made in Germany). Plants were grown in a growth chamber at 25 ± 2 °C with a 16 h photoperiod under a photon flux density of 45 ± 5 μmol m^−2^s^−1^ provided by daylight fluorescent tubes (TLD 36W/830, Philips), as indicated in De la Rubia et al. (2021) [47].

### 4.2. Bacterial Strain and Growth Conditions

*Pseudomonas syringae* van Hall 1902, CECT321 (Pph) was grown for two days at 30 °C on liquid King’s B (KB) medium at 220 rpm. For infection experiments, a final concentration of 10^8^ CFU/mL Pph was used. The Pph solution for inoculation was prepared by removing the media by centrifugation and resuspending in the same volume of sterile water, as indicated in De la Rubia et al. (2021) [47].

### 4.3. Elicitation, Pathogen Inoculation, and Sample Preparation

Leaves from common bean plants at V1 stage (with two cotyledonary leaves expanded, but not the true leaves developed) were sprayed with 2 mL per leaf of 100 μM 2,6-dichloropyridine-4-carboxylic acid (INA, Alfa Aesar, LOT: 10160271) (INA) or sterile water (Mock), as described in Martínez-Aguilar et al. [49]. After 7 days, some plants previously treated or not with INA were sprayed with 2 mL of the Pph solution on foliage leaves, resulting in Mock + Pph and INA + Pph treatments. All plants were grown for 7 more days and then the foliage leaves of 10 plants per treatment were collected and homogenized with liquid nitrogen to form a pool for each condition. All pools were stored at −80 °C until their use. Three complete independent experiments (*n* = 3) were performed (see Appendix A).

### 4.4. Reactive Oxygen Species Detection

The ROS production was determined in V1 plants leaf disks using the luminol assay [100] in a Multi-Detection Microplate Reader Synergy HT (BioTek). Leaf disks were transferred to a multi-well plate (see Appendix A), containing 200 μL water or 100 μM INA, and they were incubated overnight. Afterwards, the previous solution was removed, and 100 μL per well of a solution containing 20 μM luminol L-012 (Wako) and 100 μg/mL peroxidase from horseradish type VI.A (Sigma, P6782) were added, and incubated in the dark for 30 min. Then, the reactions were started by adding 100 μL of water (Mock condition) as negative control, 100 μL of 2 μM flg22 (flg22 condition) as positive control, 100 μL of 200 μM INA (INA condition), and 100 μL of 2 μM flg22 to those disks previously incubated overnight with 100 μM INA (preINA + flg22). The ROS production, measured as relative luminescent units (RLUs), was measured over 90 min, with an integration time of 0.6 s. Data shown represent mean ± SE from one representative experiment of three independent ones performed with similar results (see Appendix A). The total areas under the kinetic curves were integrated, and the resulting values were statistically analyzed by One way ANOVA (*p* < 0.05), post hoc Tukey test.

### 4.5. Cell Wall Isolation

The plant material (a pool of foliage leaves of 10 plants for each replicate) was powdered by homogenizing the samples in liquid nitrogen with a mortar and pestle. Three g of fresh weight powder were then treated with 50 mL 70% ethanol (*v*/*v*) for 24 h, centrifuged to remove the supernatant (×2), and then incubated with 50 mL 80% acetone (*v*/*v*) for 24 h (×2). The insoluble residues were incubated with 50 mL 2.5 U mL^−1^ α-amylase (Sigma type VI-A) in 0.01 M phosphate buffer pH 7.0 for 24 h at 37 °C (×2). Then, the remaining pellets were treated with 50 mL phenol-acetic acid-water (2:1:1 *v*/*v*/*v*) for 16 h at room temperature, with a change of solvents after 8 h of incubation. Finally, 50 mL 70% ethanol (*v*/*v*) (×3) and 50 mL 100% acetone (*v*/*v*) (×3) were sequentially added to wash the samples, and the final air-dried residues were considered CWs [61].

### 4.6. FTIR Spectroscopy

Crude CWs were used to obtain the FTIR spectra using a Jasco 4700 instrument (Tokyo, Japan). The average FTIR spectra (*n* = 10), from 800 to 1800 cm^−1^, were normalized and baseline-corrected with an ATR module of 4 cm^−1^ resolution, using Spectra Manager version 2 (2016) software by Jasco corporation (Tokyo, Japan). A Principal Component Analysis was carried out from normalized spectra with R [101], and visualized with the FactoMineR package [102].

### 4.7. Sequential Polysaccharide Extraction

The sequential polysaccharide extraction followed the protocol previously described by Rebaque et al. (2017) [61] with slight modifications. Crude CWs (10 mg CW/mL) were treated with cyclohexane-trans-1,2-diamine-*N,N,N′,N′*,-tetraacetic acid sodium salt (CDTA) at pH 6.5 for 8 h. After centrifugation, the pellets were washed with distilled water. The residues obtained were treated with 0.05 M Na_2_CO_3_ + 0.02 M NaBH_4_ and washed with distilled water, to obtain the Na_2_CO_3_ fractions. The residues obtained after centrifugation were then incubated in 0.1 N KOH + 20 mM NaBH_4_ for 8 h, and washed with distilled water, obtaining the KI fractions. For the KII fractions, the remaining pellets were treated with 4 N KOH + 20 mM NaBH_4_ for 8 h and washed with distilled water. KI and KII fractions were acidified to pH 5.0 with pure glacial acetic acid, and after that, the samples were agitated and centrifugated. Water was added to the pellets obtained after KII fraction, and then were acidified to pH 5 by adding pure glacial acetic acid. The extracts and their respective washings were combined. After its centrifugation, the supernatants were collected to form the Supernatant-Cellulose Residue (SnCR) fractions. The remaining residues were hydrolyzed with 3 mL 2N trifluoroacetic acid (TFA) for 2.5 h at 120 °C, centrifuged, and clarified supernatant was referred as TFA fraction.

### 4.8. Cell Wall Sugar Content Analysis

The cellulose content of CWs was quantified by the Updegraff method [103], under the hydrolytic conditions described by Saeman, [104], using glucose as standard.

The total sugars and uronic acids were determined by the phenol-sulfuric acid method [105], and the m-hydroxydiphenyl method [106], using D-glucose and galacturonic acid as reference, respectively. For the neutral sugar estimation, the values for total sugars and uronic acids were subtracted.

For the neutral sugar composition, samples from each fraction were hydrolyzed with 2 N TFA for 1 h at 121 °C, which resulted in monosaccharides that were derivatized to alditol acetates following the method described by Albersheim [107]. Furthermore, the alditol acetates were quantified by gas chromatography (GC) using a Perkin-Elmer equipment with a flame ionization detector (GC-FID), using a Supelco SP-2330 column and a Perkin-Elmer GC-FID, as described in Rebaque et al. (2017) [61]. Inositol was used as internal control, and monosaccharides L(−)rhamnose (Merck), L(−)fucose (Sigma), L(+)arabinose (Merck), D(+)xylose (Merck), D(+)mannose (Merck), D(+)galactose (Merck), and D(+)glucose (Panreac) as standard markers.

### 4.9. Gel Permeation Chromatography

The CW polysaccharides were size-fractionated by using a sepharose CL-4B column in a 120 mL bed-volume (1.5 cm diameter) column in pyridine/acetic acid/water (1/1/23 *v*/*v*/*v*) at 0.3 mL/min. The column was calibrated with sucrose and different commercial dextrans of known relative average molecular weight (M_w_) as described by Kerr and Fry [65]. By using the *K*_av(1/2)_ method [65], a calibration curve was obtained (log M_w_ = −5.333 *K*_av(1/2)_ + 8.103). V_0_ and V_i_ (*K*_av_ 0 and 1) were defined by dextran blue (2000 kDa) and sucrose, respectively. Then, the polysaccharides were loaded into the column at a concentration of 100 μg/mL, and the total sugars were estimated for each fraction. The nominal M_w_ for each fraction was determined (nominal rather than absolute due to the conformational differences between dextran standards and CW polysaccharides).

### 4.10. Immunodot Assays

IDAs were carried out as described by García-Angulo et al. (2006) [89]. The reference compounds were commercial pectin (P41) and arabic gum. Monoclonal antibodies (mAbs) LM1, LM2, JIM5, JIM7, LM5, LM6, LM11, and LM15 were used with mung bean as standard. The reference compounds, as well as samples of fractions, were diluted 1/5, five times. Then aliquots of 1 μL from each dilution were spotted on nitrocellulose membranes. After drying, the membranes were blocked with 0.14 M NaCl, 2.7 mM KCl, 7.8° mM Na_2_HPO_4_.12H_2_O and 1.5° mM KH_2_PO_4_, pH 7.2 (PBS), with 4% fat-free milk powder, during 1.5 h at room temperature and incubated in primary antibody (hybridoma supernatants diluted 1/10). After washing the membranes, they were incubated in secondary antibody (antirat horseradish peroxidase conjugate, Sigma) diluted 1/1000, during 1.5 h at room temperature. The color was developed with 25 mL deionized water, 5 mL methanol with 10 mg mL^−1^ 4--chloro--1--naphtol, 30 µL 6% (*v*/*v*) H_2_O_2_, and stopped by washing the membranes. A value ranging from 0 to 5 was assigned depending on the number of colored spots that were shown after developing the membranes as described in De Castro et al. (2014) [108]. As it corresponded with dilutions, this was used to establish a semiquantitative scale which was processed in a Heatmap, and clustering was performed with R [101].

### 4.11. Cell Wall Degradability

The CW degradability was assayed on 5 mg of crude CWs hydrolyzed with a cocktail of hydrolytic enzymes, containing 1% macerozyme (R10, from *Rhizopus* sp., Duchefa, EC number 232-885-6), 1% driselase (from *Basidiomicetes*, Sigma, EC number 286-055-3), 1% cellulase (Onozuka R10, from *Trichoderma viride*, Phytotechnology Laboratories, EC number 3.2.1.4), and 1% endo-polygalacturonase (M2, from *Aspergillus aculeatus*, Megazyme, EC number 3.2.1.15), in 20 mM sodium acetate (pH 4.8) (Fornalé et al., 2012). Samples were incubated at 37 °C by shaking. Aliquots of the CW hydrolysate were collected at different times, and 10 μL of 99% formic acid (Panreac) was added to stop the enzymatic digestion. The total sugars were quantified by the phenol-sulfuric method [105].

### 4.12. Statistical Analyses and Software

Results were represented using GraphPad Prism 6, GraphPad Software (La Jolla, California USA; www.graphpad.com), while the statistical analyses were performed with SPSS software (IBM Corp. Released 2017. IBM SPSS Statistics for Windows, Version 25.0. Armonk, NY: IBM Corp). Data normality was checked firstly by Kolmogorov-Smirnov test. The ROS and CW polysaccharide and monosaccharide quantification were analyzed by one-way ANOVA with a Tukey post-test to evaluate the differences between treatments. Data on digestibility were analyzed by a two-way ANOVA with a Tukey post-test. Results were significantly different considering *p <* 0.05. As mentioned above, the PCA and Heatmap were analyzed with R [101].

Foliar lesion areas were quantified using imageJ software [109].

## Figures and Tables

**Figure 1 plants-10-01514-f001:**
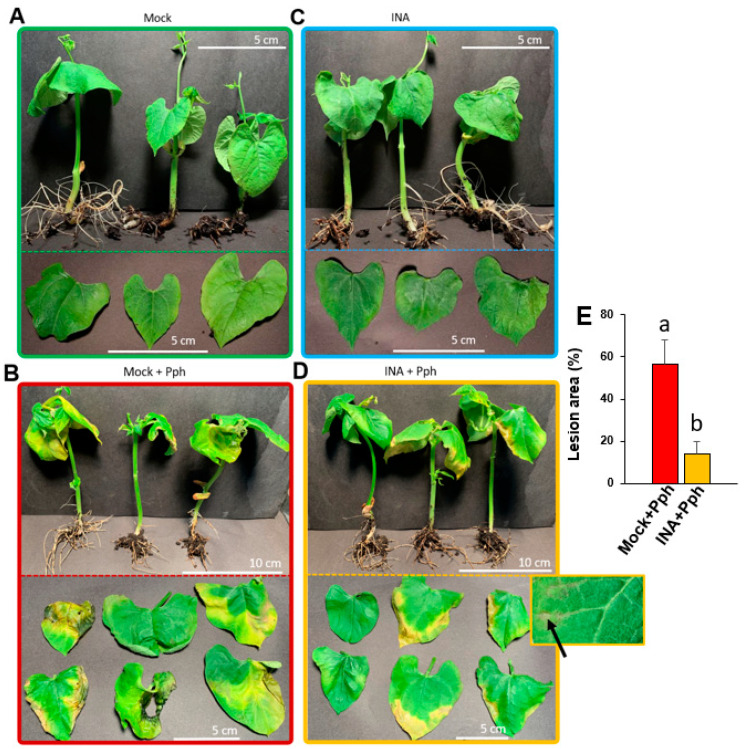
Phenotypic damage caused by the virulent bacteria *P. syringae* pv. *phaseolicola* (Pph) in common bean plants. Entire plants and extended leaves are shown to compare symptoms in Mock (**A**,**C**) and INA-pre-treated plants (**B**,**D**), non- (**A**,**B**) or Pph-inoculated (**C**,**D**). In panel D, a detail of the observed necrotic spots indicative of hypersensitive response (pointed by an arrow) is shown. (**E**) Quantification of the lesion area. Data represent mean ± SE (*n* = 6) and statistically differences, indicated by letters, were achieved according to *t*-Student test (*p* < 0.05). Pictures correspond to one experiment representative of three independent ones with similar results.

**Figure 2 plants-10-01514-f002:**
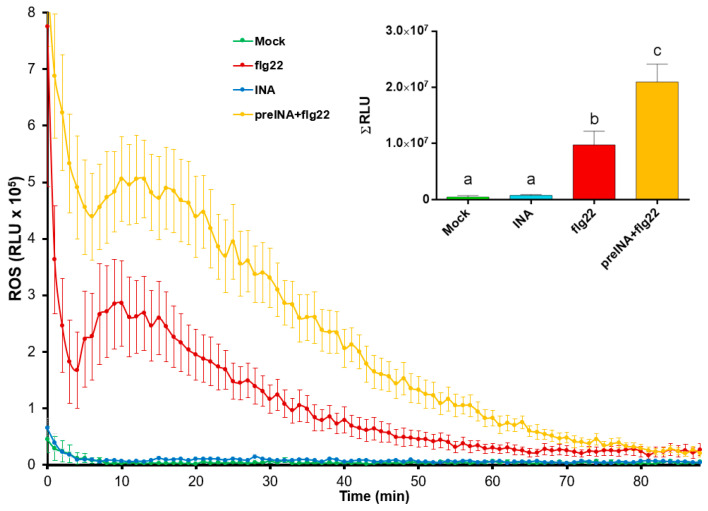
Reactive oxygen species (ROS) production in bean leaf-disks subjected to different treatments: water (Mock, green line), 100 μM INA (INA, blue line), 1 μM flg22 (flg22, red line), or 1 μM flg22 after pre-treatment with 100 μM of INA (preINA + flg22, yellow line), and measured as relative light units (RLU) produced by Luminol reaction over the time. Total areas under the curves were integrated, and resultant values are represented at the right side of the panel. Data represent mean ± SE (*n* = 8) from one experiment representative of three independent ones with similar results. Statistically significant differences, indicated by letters, were achieved according to one-way ANOVA (*p* < 0.05), by post hoc Tukey test.

**Figure 3 plants-10-01514-f003:**
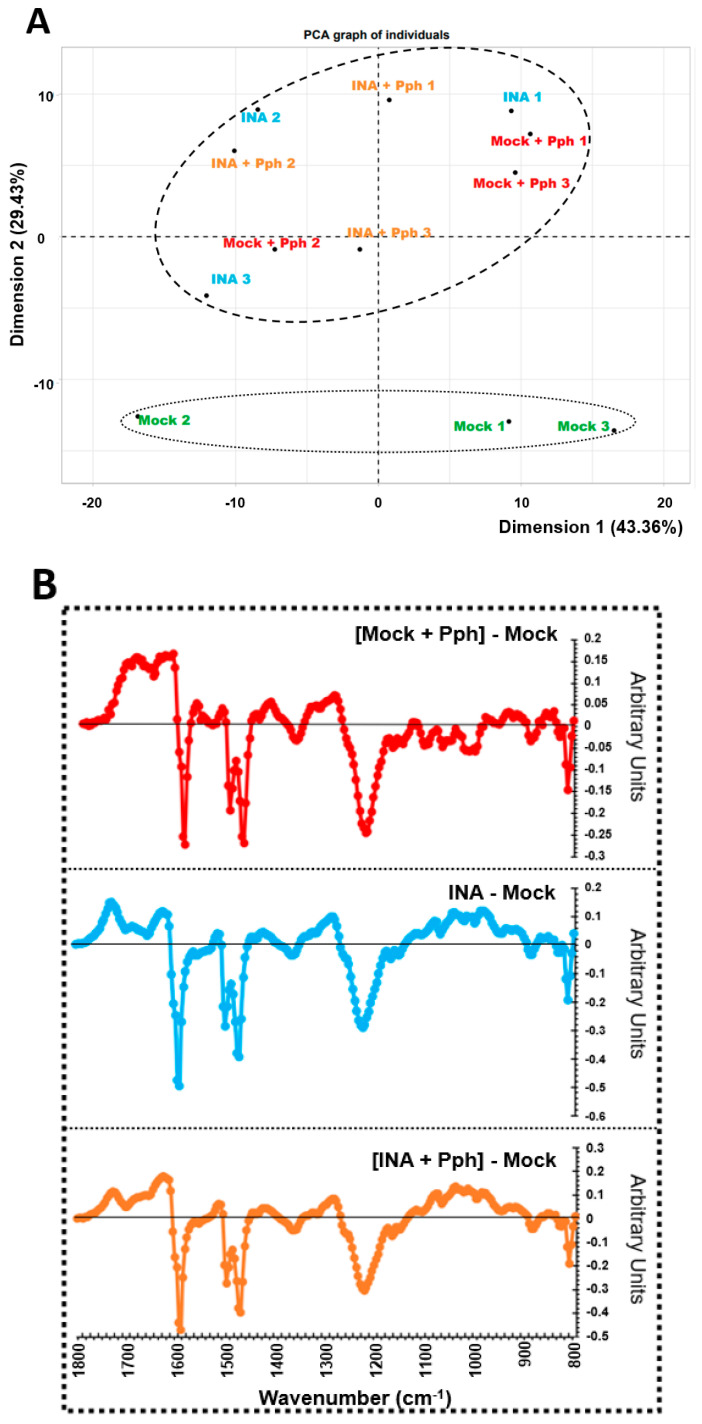
Cell wall fingerprinting. (**A**) Principal Component Analysis obtained from FTIR spectra of crude cell walls extracted from leaves of Mock, Mock + Pph, INA, and INA + Pph common bean plants. Each dot represents a biological replicate obtained in each of the 3 independent experiments carried out. PCA is plotted with dimensions 1 and 2, which explained the 43.36% and 29.43% of the total variance, respectively. (**B**) Average FTIR difference spectra obtained by digital subtraction of the Mock CW spectra from the CW spectra of the other treatments.

**Figure 4 plants-10-01514-f004:**
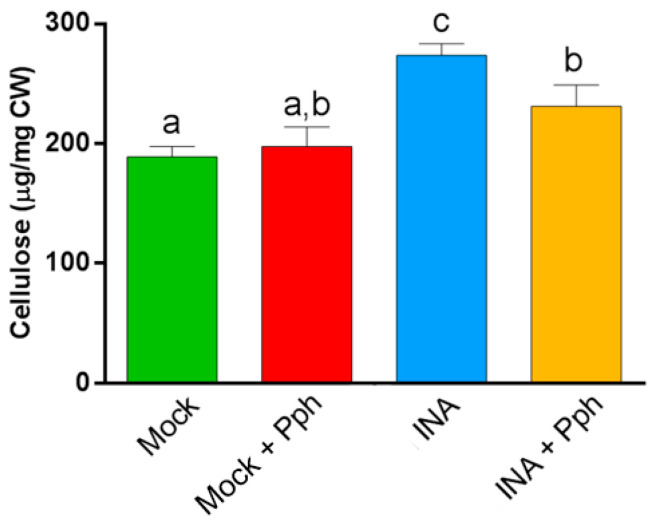
Cellulose content (µg per mg of dry weight CW) of the indicated treatments. Data represent mean ± SE (*n* = 3). Statistical analysis was carried out by one-way ANOVA where letters indicate differences by Tukey test (*p* < 0.05).

**Figure 5 plants-10-01514-f005:**
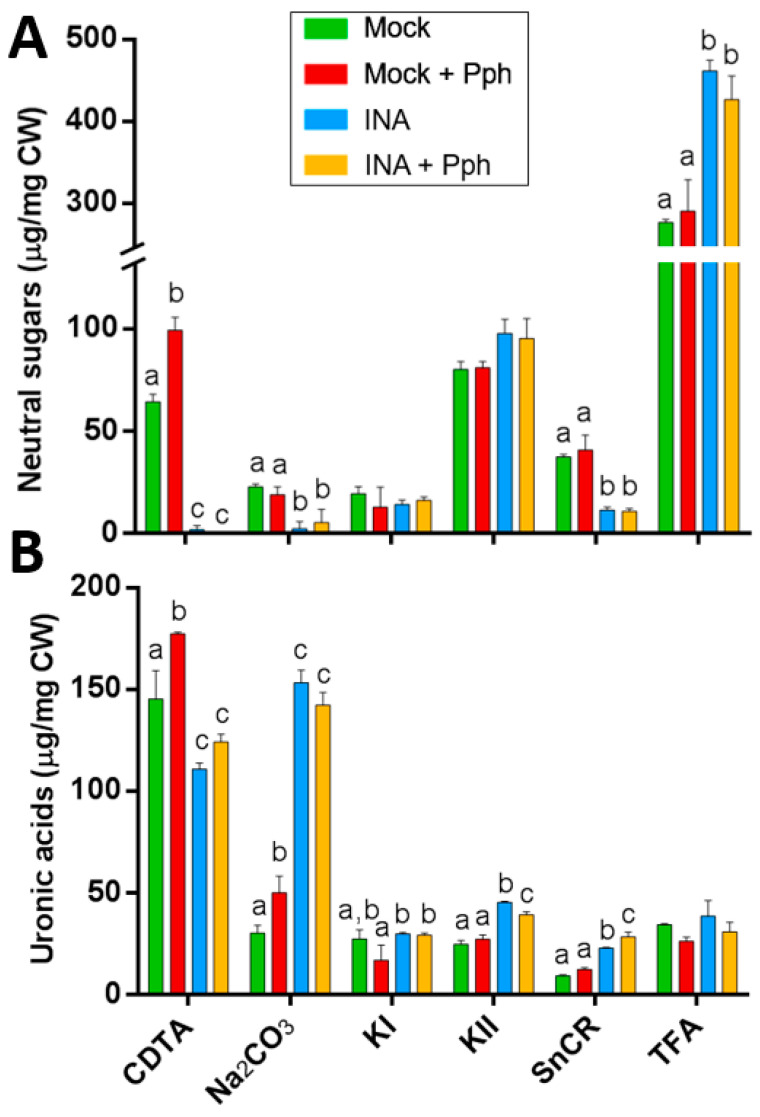
Biochemical composition of CW fractions from Mock, Mock + Pph, INA and INA + Pph inoculated common bean plants: (**A**) neutral sugars and (**B**) uronic acid content. Data represent mean ± SE (*n* = 3). Statistical analysis was carried out by one-way ANOVA where letters indicate differences by Tukey test (*p* < 0.05).

**Figure 6 plants-10-01514-f006:**
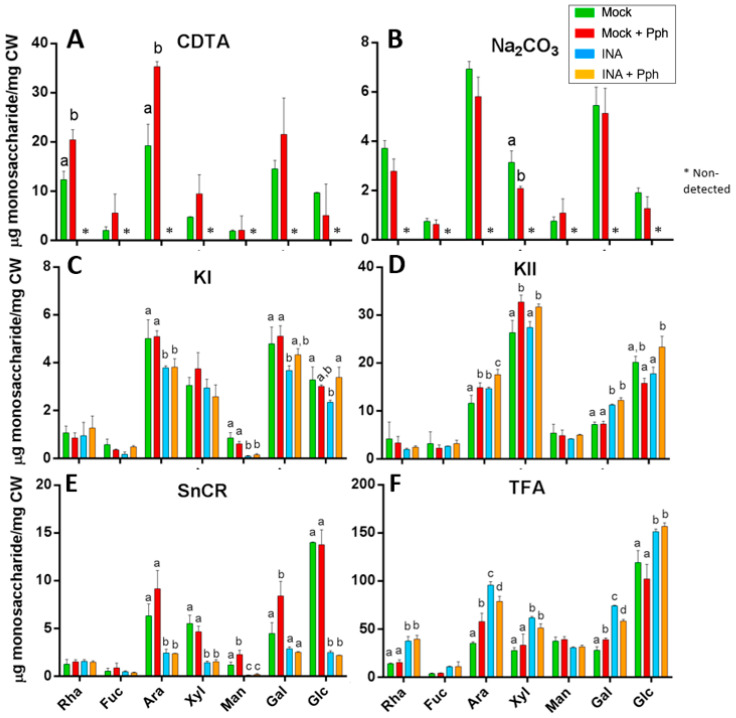
Monosaccharide analysis from different CW fractions: CDTA (**A**), Na_2_CO_3_ (**B**), KI (**C**), KII (**D**), SnCR (**E**), and TFA (**F**) of Mock, Mock + Pph, INA, and INA + Pph common bean plants. Monosaccharides are referred as Rha: rhamnose, Fuc: fucose, Ara: arabinose, Xyl: xylose, Man: mannose, Gal: galactose and Glc: glucose. Data represent mean ± SE (*n* = 3). Statistical analysis was carried out by *t*-student for Figure A and B, and by one-way ANOVA for Figure C to F where letters indicate differences by Tukey test (*p* < 0.05). Asterisks refers to non-detected monosaccharides.

**Figure 7 plants-10-01514-f007:**
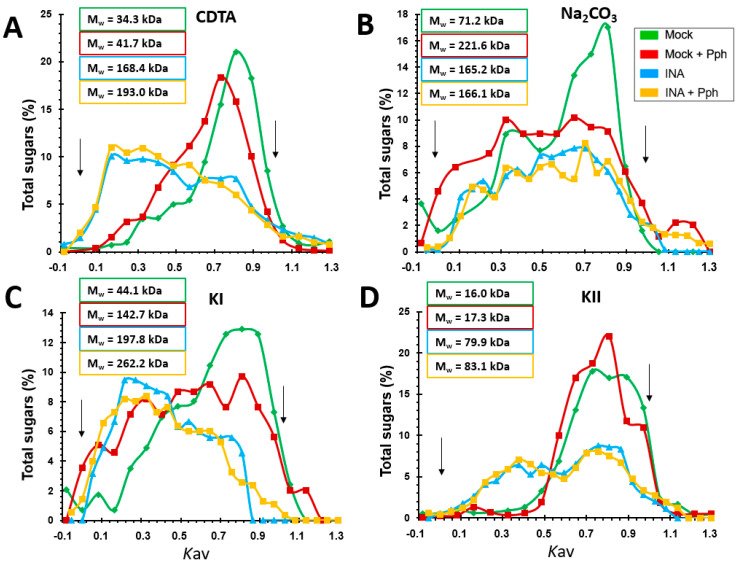
Relative mass distribution profiles of CDTA (**A**), Na_2_CO_3_ (**B**), KI (**C**), and KII (**D**) fractions extracted from CWs of Mock, Mock + Pph, INA and INA + Pph. 0 and 1 *K*av were assigned to blue dextran and sucrose markers, respectively (indicated with arrows). The average molecular weight in kDa (M_w_) obtained by *K*av(1/2) method is shown inserted in boxes.

**Figure 8 plants-10-01514-f008:**
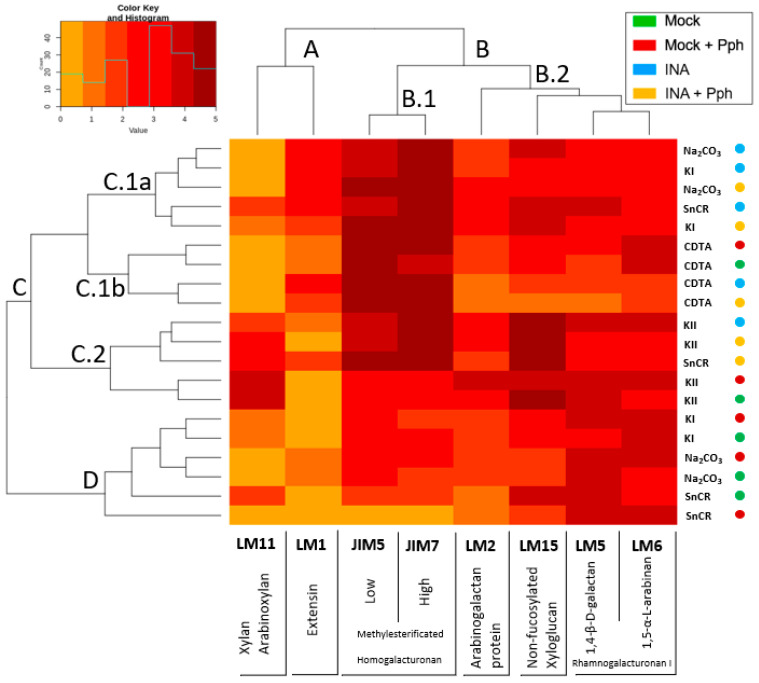
Heatmap of data obtained from IDAs of fractioned CW from Mock, Mock + Pph, INA and INA + Pph conditions. The value assigned from 0 to 5, depending on the marked dilution after development, is represented in a color key for the 5 levels. Fractions of each treatment are clustered in the horizontal axis, while monoclonal antibodies are clustered in the vertical axis. For easy visualization, the next clusters were established with respect to antibodies clusterization: cluster A, grouping LM1 and LM11; cluster B, divided into B.1, containing JIM5 and JIM7, and B.2, where LM5, LM6, LM2 and LM15 were included. Regarding fractions and treatments, the clusters obtained were: cluster C, divided into C.1a (relative to Na_2_CO_3_ and KI fraction of INA treatments), C.1b (where all CDTA fractions were located), and C.2, with KII fractions mostly; cluster D, relative to Na_2_CO_3_ and KI fraction but of Mock and Mock + Pph.

**Figure 9 plants-10-01514-f009:**
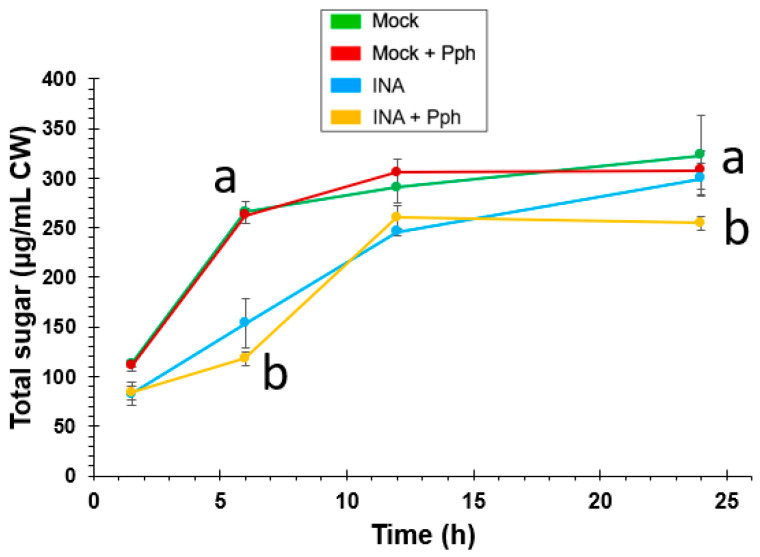
Time course of sugars released by enzymatic hydrolysis of CWs extracted from Mock, Mock + Pph, INA and INA + Pph conditions. Data represent mean ± SE (*n* = 3). Statistical analysis was carried out by two-way ANOVA where letters indicate significant differences by Tukey test (*p* < 0.05).

**Table 1 plants-10-01514-t001:** FTIR wavenumbers (cm^−1^) with highest contribution to explain Dimension 2 as Cos2 Dim2 (Figure 1) and the CW component to which they are associated.

Wavenumber (cm^−1^)	Cos2 Dim2	CW Component	Reference
808–816	8.05 × 10^−1^–6.93 × 10^−1^	Unknown	
1148	7.14 × 10^−1^	RG-I, Galactan, Xyloglucan	[55]
1152	7.98 × 10^−1^	RG-I, Galactan, Xyloglucan	[55]
1156	7.18 × 10^−1^	Arabinogalactan	[55]
1160	6.90 × 10^−1^	Cellulose	[56,57]
1164	7.39 × 10^−1^	Cellulose	[58]
1192–1196	6.43 × 10^−1^–7.26 × 10^−1^	Unknown	
1200	7.61 × 10^−1^	C = CH	[59]
1204–1228	7.98 × 10^−1^–8.12 × 10^−1^	Unknown	
1232	8.02 × 10^−1^	Pectin	[60]
1236–1240	7.72 × 10^−1^–7.26 × 10^−1^	Unknown	
1244	6.67 × 10^−1^	Pectin	[59]
1472–1596	6.61 × 10^−1^–6.14 × 10^−1^	Unknown	
1616	7.34 × 10^−1^	Free carboxyl uronic acid	[55]
1620	7.15 × 10^−1^	Free carboxyl uronic acid	[60]
1624	6.89 × 10^−1^	Free carboxyl uronic acid	[60]
1628	7.24 × 10^−1^	Free carboxyl uronic acid	[60]
1632	7.41 × 10^−1^	Phenolic ring	[60]
1636–1648	7.57 × 10^−1^–6.35 × 10^−1^	Unknown	
1720	6.09 × 10^−1^	Phenolic ester	[57,60]

## Data Availability

Not applicable.

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
