# Peer review of "Immune Priming Triggers Cell Wall Remodeling and Increased Resistance to Halo Blight Disease in Common Bean"

_plants, 2021, doi:10.3390/plants10081514_

Round 1

Reviewer 1 Report

The manuscript entitled “Immune priming triggers cell wall remodelling and increased resistance to halo blight disease in common bean” regards on the ability of INA to trigger an immune priming in Phaseolus vulgaris against Pseudomonas syringae and how this response could be mediated by a CW remodelling. The authors were able to identify some structural wall changes induced by INA that could mediate the increased resistance declared following pre-treatment. However, some experiments are required to improve these conclusions. Interestingly, different results obtained in this manuscript are consistent with findings obtained in previous manuscripts related to cell wall remodelling in other pathosystems and that are neither mentioned in the introduction nor in discussion. The work is very interesting and well-structured and the data are reported in correct English

Comments:

  • The pathogen symptoms need to be quantified (necrotic area or realtime pcr quantifying the expression of a pathogen gene).
  • Page 14 lane 193 and follow. Although it is true that the knowledge about plant CW remodelling after pathogen infection and/or immune priming is scarce some experimental evidence identified the modification of a specific cell wall polysaccharides, especially on pectin methylesterification, extensins and hemicellulose changes, as a remodelling mechanism triggered during different microbial infections. It is important to integrate in the discussion (or mention in the introduction) the newly acquired knowledge with these previous results also because in different cases they are consistent and could highlight common responses of maintaining wall integrity between plants. Plant Physiol. 2013 May; 162(1): 9–23, BMC Plant Biol. 2015; 15: 6, Front Plant Sci. 2016; 7: 630, Plant Physiol. 2017 Mar; 173(3): 1844–1863, Plant Sci 2018 Sep;274:121-128, Mol Plant Pathol. 2020 Dec; 21(12): 1620–1633.

Author Response

Please, find the answer to reviewer 1 in the attach file.

Reviewer 2 Report

The review on the publication by De La Rubia et al under the title ‘Immune priming triggers cell wall remodeling and 3 increased resistance to halo blight disease in common bean’

The publication is very detailed and well prepared. I have only minor comments to it.

Line 42-44 What about other pectic polysaccharides such as apiogalacturonan, xylogalacturonan, arabinan, galactan, arabinogalactan? I think its worth to mention that there are more of pectic polysaccharides.

Line 70 What do you mean by term plasticity? You mean to the environmental changes? What about pectins?

Author Response

Please, find the answer to reviewer 2 in the attached file
